(GIGA)byte

DATA RELEASE

# High-speed whole-genome sequencing of a Whippet: Rapid chromosome-level assembly and annotation of an extremely fast dog's genome

Marcel Nebenführ[1,2,3,*,†], David Prochotta[1,2,3,†], Alexander Ben Hamadou[1,3], Axel Janke[1,2,3], Charlotte Gerheim[1,3], Christian Betz[4], Carola Greve[1,3] and Hanno Jörn Bolz[4,*]

1 Senckenberg Biodiversity and Climate Research Centre (BiK-F), Frankfurt am Main, Germany
2 Institute for Ecology, Evolution, and Diversity, Goethe University, Frankfurt am Main, Germany
3 LOEWE-Centre for Translational Biodiversity Genomics (TBG), Frankfurt am Main, Germany
4 Bioscientia Human Genetics, Institute for Medical Diagnostics GmbH, Ingelheim, Germany

## ABSTRACT

The time required for genome sequencing and *de novo* assembly depends on the interaction between laboratory work, sequencing capacity, and the bioinformatics workflow, often constrained by external sequencing services. Bringing together academic biodiversity institutes and a medical diagnostics company with extensive sequencing capabilities, we aimed at generating a high-quality mammalian *de novo* genome in minimal time. We present the first chromosome-level genome assembly of the Whippet, using PacBio long-read high-fidelity sequencing and reference-guided scaffolding. The final assembly has a contig N50 of 55 Mbp and a scaffold N50 of 65.7 Mbp. The total assembly length is 2.47 Gbp, of which 2.43 Gpb were scaffolded into 39 chromosome-length scaffolds. Annotation using mammalian genomes and transcriptome data yielded 28,383 transcripts, 90.9% complete BUSCO genes, and identified 36.5% repeat content. Sequencing, assembling, and scaffolding the chromosome-level genome of the Whippet took less than a week, adding another high-quality reference genome to the available sequences of domestic dog breeds.

**Submitted:** 22 May 2024

\* Corresponding authors. E-mail:
marcel.nebenfuehr@senckenberg.de;
hanno.bolz@bioscientia.de

† Contributed equally.

Preprint submitted at https://doi.org/10.1101/2024.08.16.608262

**Subjects** Genetics and Genomics, Animal Genetics, Animal Physiology

# DATA DESCRIPTION

## Background information

Although recent advances in sequencing technologies made genomics more accessible to a broader scientific community, sequencing and assembling large eukaryotic genomes of non-model organisms remains challenging. Even with access to samples of sufficient quality, limited funding for sequencing, insufficient computing power, lack of sequencing technology and capacity, and the need for outsourcing can significantly increase turnaround times.

In response to the accelerating global biodiversity loss, national and international initiatives expanded genomic references for biodiversity research and conservation. The use of reference genomes in population genomics facilitates the characterization of genetic

**Figure 1.** **Project timeline with day-by-day progress description and time requirements.**
The blue line represents the contribution of the biodiversity research centre (TBG) at each step, and the red line represents the contributions made by the medical diagnostics company (Bioscientia).

diversity and adaptation through local variant enrichment, providing a basis for biodiversity assessment, conservation, and restoration [1–3].

Furthermore, non-human genomes have become increasingly helpful for the interpretation of human genetic variants and their relevance for disease [4, 5]; this particularly applies to *de novo*-assembled long-read human genomes that, in contrast to short-read genomes, allow for the identification of complex (structural) variants that would otherwise escape detection. In both biodiversity and conservation research, as well as in medical genetics, the timely generation and assembly of genomic data is of utmost importance.

In this work, an academic biodiversity institute joined forces with a medical diagnostic company with extensive sequencing capacity and experience to generate a rapid chromosome-level *de novo* genome of the Whippet. We demonstrated that streamlined laboratory and bioinformatics workflows with PacBio high-fidelity (HiFi) long-read whole-genome sequencing (LR-WGS) enable the generation of a high-quality reference genome within one week (Figure 1). Besides this proof-of-concept study of the rapid Whippet genome, our collaboration includes continuous LR-WGS of *de novo* genomes of various endangered species (including non-vertebrates and plants). Our study may serve as a paradigm for such cooperations applicable to a wide range of human and non-human genome projects, from biodiversity research to domestic animal and agricultural research.

### Sampling, DNA extraction, and sequencing

High molecular weight genomic DNA was extracted from the peripheral blood leukocytes of a three-year-old male dog (*Canis lupus familiaris*, NCBI:txid9615), a Whippet, using the PacBio Nanobind CBB kit (Pacific Biosciences, Menlo Park, CA). Blood was taken during a routine veterinary procedure, collected in EDTA-coated vials, and frozen at −20 °C. DNA concentration and DNA fragment lengths were evaluated using the Qubit dsDNA BR Assay kit on the Qubit Fluorometer (Thermo Fisher Scientific, Waltham, MA) and the Genomic DNA Screen Tape on the Agilent 4150 TapeStation system (Agilent Technologies, Santa Clara, CA), respectively. Two SMRTbell libraries were prepared according to the instructions in the SMRTbell Express Prep Kit v3.0. The final concentrations were 68 ng/µl and 76 ng/µl, with a total input of approximately 10 µg of sheared DNA per library. Annealing of sequencing primers, binding of sequencing polymerase, and purification of polymerase-bound SMRTbell complexes were performed using the Revio polymerase kit (PacBio, Menlo Park, CA, USA). The loading concentration for sequencing was 250 pM.

### Genome assembly and polishing

The two sequencing runs on a PacBio Revio® instrument of the Whippet yielded a total of ~128 Gbp of sequence data, with an average subread N50 of ~17.8 kbp. We assembled the genome using Hifiasm v0.18.8-r525 (RRID:SCR_021069) [6] with default settings and scanned the initial genome assembly for contamination using FCS-GX v0.4.0 [7].

We polished the raw assembly three times using Inspector v1.0.1 [8], which includes Flye v2.9.3 (RRID:SCR_017016) [9] for structural error correction.

After polishing, we performed reference-guided scaffolding using RagTag v2.1.0 [10] with default settings. For that, the NCBI reference genome for dogs was chosen as a reference, which is the German Shepherd dog genome (GCA_011100685.1). Next, we used TGS-GapCloser v1.2.1 (RRID:SCR_017633) [11] to close assembly gaps using three rounds of gap-filling and the Racon v1.0.5 (RRID:SCR_017642) [12] module for polishing the assembly gaps.

### Assembly quality control

We evaluated both the raw assembly and the final assembly with Merqury v1.3 (RRID:SCR_022964) [13] in combination with Meryl v1.4.1, as well as Inspector v1.0.1.

We calculated assembly contiguity statistics of the final genome using Inspector. We then performed a gene set completeness analysis of the Whippet genome and other available dog genomes for comparison using BUSCO v5.4.22 (RRID:SCR_015008) [14] with the provided Carnivora orthologous genes database (carnivora_odb10).

We mapped the reads back to the genome using minimap2 v2.28 (RRID:SCR_018550) [15] with the '-ax map-hifi' option to output a SAM file. The SAM file was sorted and converted to a BAM file with samtools v1.20 (RRID:SCR_002105) [16]. We then marked duplicate sequences with sambamba v1.0.1 (RRID:SCR_024328) [17] and analyzed mapping statistics with QualiMap v2.2.1 (RRID:SCR_001209) [18].

The initial assembly, based on the HiFi long reads only, already recovered 36 (of 39) chromosome-sized scaffolds. This further demonstrates how effective and therefore invaluable accurate long-read sequencing has become in mammalian genomics.

The final polished, scaffolded, and gap-closed assembly consists of 148 scaffolds with a total length of 2.47 Gbp, while 2.43 Gbp were placed into 39 chromosome-sized scaffolds

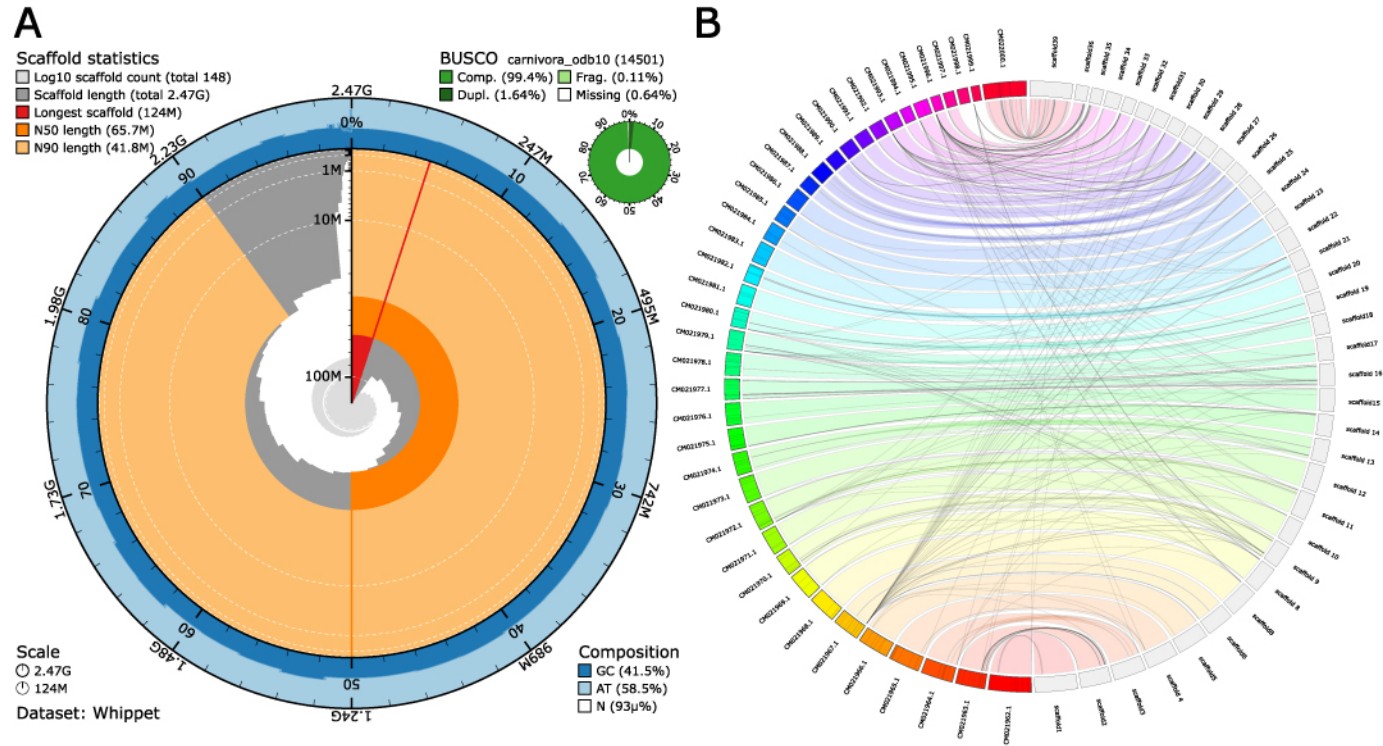

**Figure 2.** (A) Snailplot based on the polished and reference-scaffolded assembly showing BUSCO gene completeness results and basic assembly statistics. (B) Whole-genome synteny between the chromosome-level assembly of the German Shepherd and our chromosome-level assembly of the Whippet.

**Table 1.** Basic assembly statistics, Inspector and Merqury quality values of the raw and final Whippet assembly.

|  | **Raw assembly** | **Final assembly** |
|---|---|---|
| Total length (bp) | 2,472,523,283 | 2,472,512,883 |
| No. of contigs/scaffolds | 172 | 148 |
| N50 | 54,692,731 | 65,741,809 |
| L50 | 17 | 15 |
| Longest contig | 123,386,781 | 124,133,413 |
| Inspector QV | 50.2 | 53.5 |
| Inspector mapping rate (%) | 100 | 100 |
| Merqury QV | 65.6 | 68.1 |
| Merqury completeness | 97.87% | 97.87% |

(Figure 2, A and B; Table 1). Gene completeness analysis of the genome based on BUSCO's Carnivora dataset identified 14,170 complete single-copy orthologous sequences, corresponding to 97.72% completeness and 77 (0.53%) missing genes (Figure 2A).

Both Inspector and Merqury, coupled with the high BUSCO score, indicated a highly contiguous, accurate, and complete genome. In addition, available dog assemblies were downloaded to compare the BUSCO completeness across published reference genomes (Figure 3, Table 2). In addition, the qualimap report for the resulting BAM file showed 99.98% of mapped reads (8,332,294), a mean mapping quality of 49.7, a mean coverage of 55.1X, and an error rate of 0.0057.

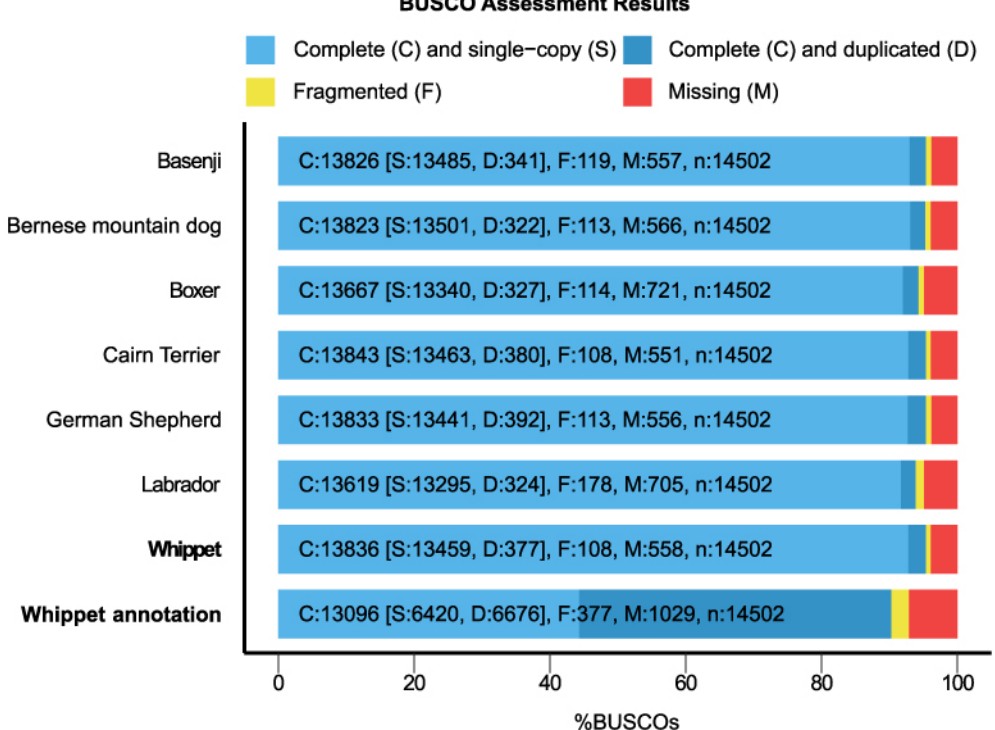

**Figure 3.** Comparison of BUSCO completeness statistics based on the carnivora database between our final Whippet assembly and annotation, and other available dog assemblies.
Complete single-copy genes are shaded light blue, and complete duplicated sequences are shaded dark blue; fragmented genes are shaded yellow, and missing sequences are shaded red. The numbers of complete single copy (S), complete duplicated (D), fragmented (F), and missing genes (M) for the respective genome are shown in each column. The total number of genes in the BUSCO carnivora library is denoted as *n*.

**Table 2.** Available genome data used for comparison.

| Species | Accession number |
|---|---|
| German Shepherd (*Canis lupus familiaris*) | GCA_011100685.1 |
| Labrador Retriever (*Canis lupus familiaris*) | GCA_012044875.1 |
| Basenji (*Canis lupus familiaris*) | GCA_013276365.2 |
| Cairn Terrier (*Canis lupus familiaris*) | GCA_031010295.1 |
| Bernese Mountain Dog (*Canis lupus familiaris*) | GCA_031010765.1 |
| Boxer (*Canis lupus familiaris*) | GCF_000002285.5 |
| Cat (*Felis catus*) | GCF_018350175.1 |
| Dingo (*Canis lupus dingo*) | GCF_003254725.2 |
| Mouse (*Mus musculus*) | GCF_000001635.27 |
| Human (*Homo sapiens*) | GCF_009914755.1 |

## Heterozygosity

To calculate genome-wide heterozygosity in the Whippet genome, we first mapped the reads used for assembly to the genome and marked duplicates using sambamba v1.0.0, then counted base-depth at all sites using sambamba. We then estimated the Site Frequency Spectrum (SFS) with ANGSD v0.940 (RRID:SCR_021865) [19] and used the output files to run realSFS (RRID:SCR_002493) with 200 bootstrap replicates to calculate the folded SFS. From this, we calculated the heterozygosity by dividing the heterozygous sites by the sum of the

**Table 3.** Repeat content of the Whippet genome assembly. Class: class of the repetitive regions. Count: number of occurrences of the repetitive region. bpMasked: number of base pairs masked; %masked: percentage of base pairs masked. LINE: Long Interspersed Nuclear Elements (including retroposons); LTR: Long Terminal Repeat elements (including retroposons); SINE: Short Interspersed Nuclear Elements; RC: Rolling Circle. In total, 902,477,158 bp were masked, corresponding to 36.5% of the genome.

| Class | Count | bpMasked | %masked |
|---|---|---|---|
| SINEs | 478,438 | 65,159,195 | 2.64 |
| LINEs | 1,796,370 | 502,326,946 | 20.32 |
| LTR | 354,096 | 90,945,042 | 3.68 |
| DNA transposons | 300,163 | 49,080,228 | 1.99 |
| Rolling-circles | 1,568 | 78,131 | 0.00 |
| Unclassified | 525,739 | 132,765,950 | 5.37 |
| Small RNA | 61,933 | 5,023,647 | 0.2 |
| Satellites | 14,007 | 2,268,618 | 0.09 |
| Simple repeats | 870,502 | 48,791,971 | 1.97 |
| Low complexity | 109,450 | 6,037,430 | 0.24 |

homozygous and heterozygous sites. In line with other dog genomes, the heterozygosity was 0.09%. Our heterozygosity analysis delivers standard baseline data for a *de novo* sequenced genome and allows a first glimpse into the genetic diversity of Whippets. However, it cannot be representative of Whippets in general.

## GENOME ANNOTATION

### Repeat annotation

To annotate repetitive regions in the Whippet genome, we used RepeatModeler v2.1 (RRID:SCR_015027) [20] to create a *de novo* repeat library for our assembly. Next, we used RepeatMasker v4.1.6 (RRID:SCR_012954) [21] to hard-mask repeats based on the modeled repeats. Our analyses identified 36.5% of repeats in the genome, of which the majority consisted of long interspersed nuclear elements (LINEs) (20.32%) and long terminal repeat (LTR) elements (3.68%). In addition, 5.37% of unclassified elements were identified (Table 3).

### Gene annotation

Gene annotation was performed on the unmasked assembly using GeMoMa v1.9 (RRID:SCR_017646) [22]. Therefore, available annotated high-quality genomes of dogs and other mammals were used as references to identify genes (Table 2). The final gene annotation resulted in 28,383 transcripts. In addition, our BUSCO analysis identified 90.9% complete BUSCOs, suggesting a high annotation completeness (Figure 3).

### Myostatin

Since a founder mutation in myostatin (*MSTN*) has been reported in Whippet dogs [23], we analyzed our Whippet genome for this mutation by comparing its *MSTN* sequences to the wild-type references of *Canis lupus familiaris* (AY367768) and found no mutation in the sequence (Figure 4). We first identified the position of the gene within the genome with MMseqs2 (RRID:SCR_022962) [24] and then visualized the region in IGV (RRID:SCR_011793) [25] to check for mutations in the aligned reads. Hence, the analysis of our Whippet genome's *MSTN* sequence primarily exemplifies a possible application of available high-quality genomes, especially for breeding purposes, but is not representative of Whippets in general.

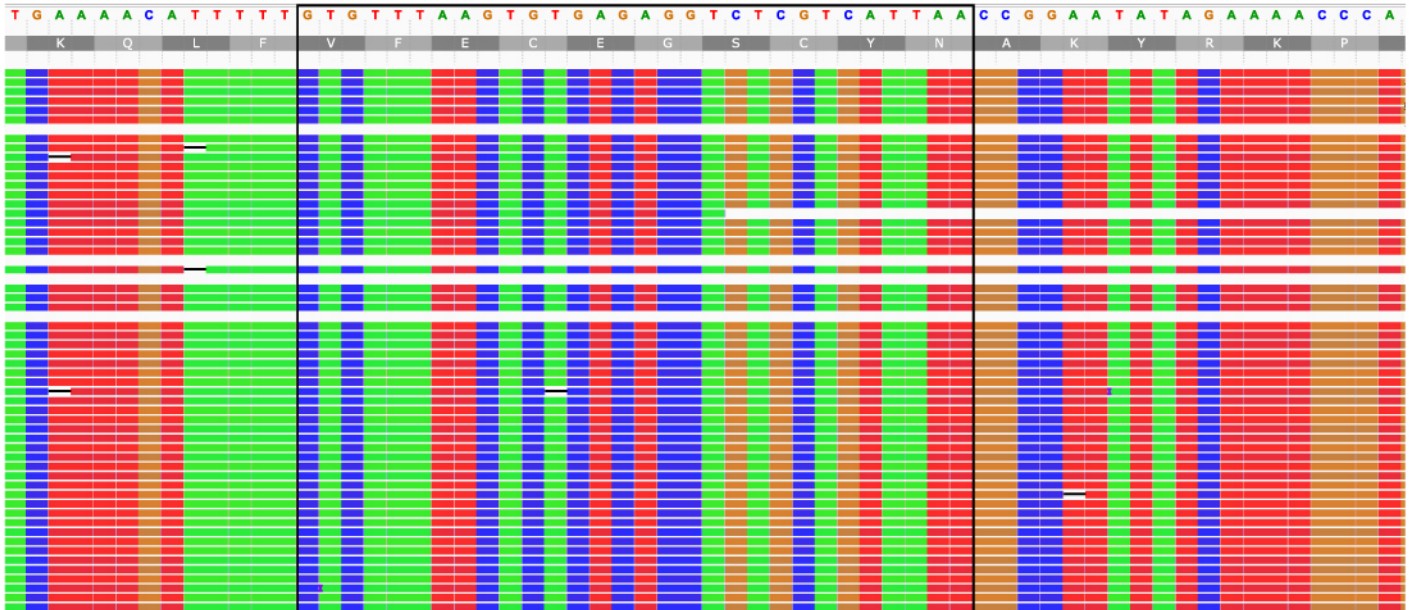

**Figure 4. Read alignment of the myostatin gene, *MSTN*, in the sequenced Whippet genome.**
The black box indicates the *MSTN* sequence of interest, as used by Mosher *et al.* [23].

## CONCLUSION

In our proof-of-concept study, we show that teaming up a medical diagnostics company with a biodiversity research institute may deliver extremely rapid *de novo*-assembled HiFi long-read genomes. This was possible through close and streamlined time management and collaboration, including all required participants for a genome project, namely a veterinarian, laboratory facilities, the sequencing facility, and the bioinformatics unit.

## DATA AVAILABILITY

All raw data generated in this study are accessible at GenBank under *BioProject* PRJNA1114051. Annotation, results files, and other data are available in the GigaDB repository [26].

## ABBREVIATIONS

HiFi, High Fidelity; LINE, long interspersed nuclear element; LR-WGS, long-read whole-genome sequencing; LTR, long terminal repeat; MSTN, myostatin; SFS, Site Frequency Spectrum.

## DECLARATIONS

### Ethics approval and consent to participate
Not applicable.

### Competing interests
CB and HJB are employees of Bioscientia, which is part of a publicly traded diagnostic company. The authors declare that they have no competing interests.

## Author contributions

ABH and ChG performed the DNA extraction and the library preparation. CB supervised genome sequencing and data transfer. DP and MN assembled and analyzed the genomes, and conducted the downstream analyses. CG, MN, DP, and HJB jointly supervised the project and wrote the manuscript with input from ABH, AJ, ChG, and CB. All authors read and approved the final manuscript before submission.

## Funding

The contribution of the Centre for Translational Biodiversity Genomics (LOEWE-TBG) to the study was funded by the Hessen State Ministry of Higher Education, Research and the Arts (LOEWE/1/10/519/03/03.001(0014)/52).

## Acknowledgements

We thank Kristina Grund, Kleintierpraxis Auringen, Germany, for veterinary care, and Ursula Wollscheid, Inge Lischewski, Lea Arndt and Anna Linck for technical assistance. We also thank Christian Decker and Sebastian Görges for bioinformatics support.

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
