## [Editor Report]

Editor’s AssessmentThis Data Release paper presents the genome of the whippet breed of dog. Demonstrating a streamlined laboratory and bioinformatics workflows with PacBio HiFi long-read whole-genome sequencing that enables the generation of a high-quality reference genome within one week. The genome study being a collaboration between an academic biodiversity institute and a medical diagnostic company. The presented method of working and workflow providing examples that can be used for a wide range of future human and non-human genome projects. The final is 2.47 Gbp assembly being of high quality - with a contig N50 of 55 Mbp and a scaffold N50 of 65.7 Mbp. This reference being scaffolded into 39 chromosome-length scaffolds and the annotation resulting in 28,383 transcripts. The results also looked at the Myostatin gene which can be used for breeding purposes, as these heterozygous animals can have an advantage in dog races. The reviewers making the authors clarify this part a little better with additional results. Overall this study demonstrating how rapidly animal genome research can be carried out through close and streamlined time management and collaboration.Editor’s AssessmentThis Data Release paper presents the genome of the whippet breed of dog. Demonstrating a streamlined laboratory and bioinformatics workflows with PacBio HiFi long-read whole-genome sequencing that enables the generation of a high-quality reference genome within one week. The genome study being a collaboration between an academic biodiversity institute and a medical diagnostic company. The presented method of working and workflow providing examples that can be used for a wide range of future human and non-human genome projects. The final is 2.47 Gbp assembly being of high quality - with a contig N50 of 55 Mbp and a scaffold N50 of 65.7 Mbp. This reference being scaffolded into 39 chromosome-length scaffolds and the annotation resulting in 28,383 transcripts. The results also looked at the Myostatin gene which can be used for breeding purposes, as these heterozygous animals can have an advantage in dog races. The reviewers making the authors clarify this part a little better with additional results. Overall this study demonstrating how rapidly animal genome research can be carried out through close and streamlined time management and collaboration.

---

## [Reviewer Report]

Indicate in the comments box below whether you are happy with the changes made or if the manuscript is unacceptable.Comments on revised manuscriptI saw the authors made some changes which has improved the mauscript, but I still have some minor comments which I think the authors should be further consider before it could be published. 1. I recommend the authors further remove key words from the current version, so redundent with the title, at least delete the biodiversity conservation, because this manuscript has nothing to do with this. 2. The author persist to retain the part of Heterozygosity, but this contribution to conservation is very very insignificant. The authors should at least clarify the limitation for it. 3. The author persist to retain the part of Myostatin. I agree that this is helpful to breeding but one individual could provide very very little solid genetic information for it. The authors should also clarify the limitation.

---

## [Reviewer Report]

Reviewer name and names of any other individual's who aided in reviewer Tianming LanDo you understand and agree to our policy of having open and named reviews, and having your review included with the published papers. (If no, please inform the editor that you cannot review this manuscript.)YesIs the language of sufficient quality?YesPlease add additional comments on language quality to clarify if needed
Are all data available and do they match the descriptions in the paper? YesAdditional CommentsAre the data and metadata consistent with relevant minimum information or reporting standards? See GigaDB checklists for examples <a href="http://gigadb.org/site/guide" target="_blank">http://gigadb.org/site/guide</a>YesAdditional CommentsIs the data acquisition clear, complete and methodologically sound?YesAdditional CommentsIs there sufficient detail in the methods and data-processing steps to allow reproduction?YesAdditional CommentsIs there sufficient data validation and statistical analyses of data quality? Not my area of expertiseAdditional CommentsIs the validation suitable for this type of data?YesAdditional CommentsIs there sufficient information for others to reuse this dataset or integrate it with other data?YesAdditional CommentsAny Additional Overall Comments to the AuthorThe authors provided an example of High-speed strategy for whole-genome sequencing, genome assembly and annotation for species and take an example with the Whippet dog. This is a very novel idea under the genomic era with plummeting sequencing cost, fast accumulated sequencing data but shortage of computing resources. The authors also provide a very high-quality reference genome for the Whippet dog species with very good contiguity, accuracy and completeness. However, I have several concerns need the authors to further consider before it could be published at the journal of Gigatyte. Q1. There are too many keywords. Can the authors reduce a few? Biodiversity conservation, Comparative genomics, and evolutionary biology does not make sense in this manuscript. Q2. The authors performed reference-guided scaffolding analysis with the German Shepherd dog genome (GCA_011100685.1) as reference. Better if the authors explain why they selected this genome as the reference as there are several published dog genomes? Q3.The part of Heterozygosity make no sense to this manuscript unless there is a reasonable connection with other parts, because the dog is not a threatened species and also not a very special breed facing extensive inbreeding abd accumulation of deleterious mutations? Q4. The part of Myostatin doesn’t make sense to me, as I have read the paper the author cited and found that not all Whippet have this mutation? They sequenced 22 individuals, and 4 individuals are homozygous (-/-), 5 are heterozygous (mh/+) and the rest are homozygous (+/+). So you can always have a result by checking this mutation, but make no sense. Furthermore, one individual can hardly represent a species or a population? At the beginning of this paragraph, please change “Since” to “Since”. Q5. I think the most important find in this manuscript is how the authors finished a high-quality genome within a very short-term working. I suggest the authors remove the descriptions of Heterozygosity and Myostatin, but added a paragraph to tell readers the basic needs or standards for such a short-term work for genome assembly for a genome of something like dog. Just a suggestion, but I think would be better to improve the manuscript.RecommendationMajor Revision

---

## [Reviewer Report]

Reviewer name and names of any other individual's who aided in reviewer Xiaobo WangDo you understand and agree to our policy of having open and named reviews, and having your review included with the published papers. (If no, please inform the editor that you cannot review this manuscript.)YesIs the language of sufficient quality?YesPlease add additional comments on language quality to clarify if needed
Are all data available and do they match the descriptions in the paper? NoAdditional CommentsNo link to the relevant data in GigaDB was provided.Are the data and metadata consistent with relevant minimum information or reporting standards? See GigaDB checklists for examples <a href="http://gigadb.org/site/guide" target="_blank">http://gigadb.org/site/guide</a>NoAdditional CommentsNo link to the relevant data in GigaDB was provided.Is the data acquisition clear, complete and methodologically sound?YesAdditional CommentsIs there sufficient detail in the methods and data-processing steps to allow reproduction?YesAdditional CommentsIs there sufficient data validation and statistical analyses of data quality? YesAdditional CommentsIs the validation suitable for this type of data?YesAdditional CommentsIs there sufficient information for others to reuse this dataset or integrate it with other data?YesAdditional CommentsAny Additional Overall Comments to the AuthorThis study outlines an approach to expedite the sequencing and de novo assembly of genomes by leveraging collaboration between academic biodiversity institutes and a medical diagnostics company with advanced sequencing capabilities. The primary focus was on generating a high-quality de novo genome of the Whippet, a fast dog breed, within an accelerated timeframe. Below are some specific comments I would like to highlight. 1. The authors mentioned the use of QUAST and QualiMap software tools to assess the genome of the Whippet; however, the corresponding results were not presented in the manuscript. 2. The authors' reliance solely on mammalian protein sequences for homology annotation means that unique genes specific to the Whippet remain unannotated. The discrepancy of approximately 7% between the completeness assessments of the gene set and the genome via BUSCO further underscores the incomplete nature of the gene set. To address this, I recommend integrating transcriptome data, at the very least, to incorporate de novo annotation results. This addition should enhance the comprehensiveness and accuracy of gene annotations for the Whippet genome. 3. The authors claim the absence of reported mutations in the Mstn gene but have not provided corroborating evidence, such as read alignment results from the genomic region, to verify that this is not due to assembly errors. 4. If feasible, I propose integrating second-generation sequencing to further polish the genome and elevate its quality.RecommendationMinor Revision